# Tracking down the White Plague. Chapter two: The role of endocranial abnormal blood vessel impressions and periosteal appositions in the paleopathological diagnosis of tuberculous meningitis

Olga Spekker[1]*, Michael Schultz[2], László Paja[1], Orsolya A. Váradi[1,3], Erika Molnár[1], György Pálfi[1], David R. Hunt[4]

1 Department of Biological Anthropology, University of Szeged, Szeged, Hungary, 2 Institut für Anatomie und Embryologie, Zentrum Anatomie, Universitätsmedizin Göttingen, Göttingen, Germany, 3 Department of Microbiology, University of Szeged, Szeged, Hungary, 4 Department of Anthropology, National Museum of Natural History, Smithsonian Institution, District of Columbia, Washington, D.C., United States of America

☯ These authors contributed equally to this work.
* olga.spekker@gmail.com

## Abstract

Although endocranial abnormal blood vessel impressions (ABVIs) and periosteal appositions (PAs) have been considered as paleopathological diagnostic criteria for tuberculous meningitis (TBM) based on findings of previous studies, they are not pathognomonic for tuberculosis (TB). Therefore, their utilization in the paleopathological practice can be questioned, especially in consideration that most of the previous studies were not performed on identified skeletal collections but on osteoarchaeological material and did not include statistical data analysis. To fill the aforementioned research gap, for the first time, a macroscopic investigation was conducted on identified pre-antibiotic era skeletons from the Terry Collection. A sample set of 234 individuals who died of TB (TB group) and 193 individuals who died of non-tuberculous causes (NTB group) were examined. The frequency of ABVIs and PAs, as well as other probable TB-related lesions was recorded. To determine the significance of difference (if any) in the frequencies of ABVIs and PAs between the two groups, $\chi^2$ testing of our data was performed. We found that ABVIs, PAs, and their co-occurrence with each other and with other probable TB-related lesions were more common in the TB group than in the NTB group. In addition, the $\chi^2$ comparative frequencies of ABVIs and PAs revealed a statistically significant difference between individuals who died of TB and individuals who died of NTB causes. Our findings strengthen those of previous studies that ABVIs and PAs are not specific to TBM but can be of tuberculous origin. Therefore, they do have a diagnostic value in the identification of TB in human osteoarchaeological material, especially when they simultaneously occur with other probable TB-related lesions. Their prudent utilization provides paleopathologists with a stronger basis for diagnosing TB and consequently, a more sensitive means of assessing TB frequency in past human populations.

**Data Availability Statement:** All relevant data are within the manuscript and its Supporting Information files.

**Funding:** This work was funded by the Hungarian State EötvösFellowship 2016 (77466) of the Tempus Public Foundation, the NTP-NFTÖ-16 (1116) of the Hungarian Ministry of Human Capacities & Human Capacities Grant Management Office, and the University of Szeged Open Access Fund (4765) to OS. The National Research, Development and Innovation Office (Hungary) (K 125561) provided funding for GP. The funders had no role in study design, data collection and analysis, decision to publish, or preparation of the manuscript.

**Competing interests:** The authors have declared that no competing interests exist.

# Introduction

Tuberculosis (TB), also known as the "White Plague", is one of the oldest known infectious diseases that has been afflicting humans and animals for thousands of years [1–3]: the earliest recognized and verified human cases with TB [e.g., 4–7] come from the Neolithic period. Apparently, TB remained relatively sporadic until the 1700s but, as a consequence of increased population density and unsanitary living conditions, started to reach epidemic levels during the Industrial Revolution [8–10]. From the second half of the 19th century, a number of factors (such as the general improvement in living conditions, sanitation, and nutrition) contributed to reducing the number of cases with TB in developed countries [8–9,11–12]. TB incidence rates declined even more by the introduction of the Bacillus Calmette-Guérin (BCG) vaccine and the use of antibiotics (e.g., streptomycin, isoniazid, and rifampicin) in the management of the disease [8–9,13]. The steady decrease in TB incidence in developed countries had led to predictions of the complete eradication of TB by the end of the 20th century [14]. However, fueled by the growing human immunodeficiency virus/acquired immune deficiency syndrome (HIV/AIDS) pandemic and the emergence of multidrug-resistant tuberculosis (MDR-TB), TB incidence began to increase again in the late 1980s in both developing and developed countries [2–3,8–9,13]. In 1993, TB was declared a global public health threat by the World Health Organization [15]. Despite significant advances in the global fight against TB in the last few decades, it is still one of the top ten causes of death and the leading cause of death from a single infectious agent worldwide (ranking above HIV/AIDS) [16]. TB accounts for about 10.0 million incident cases and 1.5 million deaths annually [16]. The global public health emergency presented by TB today has produced a renewed interest and funding for TB research [12,17–18]. This has included paleopathological diagnostics of TB in earlier populations to understand its etiology [18]. The paleopathological research of TB can provide invaluable data on the manifestations of TB and the effects of the disease upon human mortality and morbidity worldwide in both prehistoric and historic times [18–20].

Paleopathological diagnosis of TB principally relies on the identification of macroscopic lesions in the human skeleton that have been found to be related to different manifestations of the disease (e.g., skeletal TB, pulmonary TB and/or TB pleurisy, and TB meningitis (TBM)) through clinical study [20–21]. Since the late 20th century, a number of studies [e.g., 19,22–34] were performed on osteoarchaeological series and documented skeletal collections that have revealed a positive association between TBM and a few endocranial alteration types, i.e., granular impressions (GIs), abnormally pronounced digital impressions (APDIs), abnormal blood vessel impressions (ABVIs), and periosteal appositions (PAs).

ABVIs and PAs on the inner surface of the skull were recognized as vestiges of the inflammatory and/or hemorrhagic processes of the meninges [e.g., 19,23–26,35–36]. These non-specific bony changes can be generated by a number of infectious (e.g., non-specific and specific meningitis, including TBM) and non-infectious (e.g., trauma and scurvy) conditions. The initial stage of the hemorrhagic process involves the formation of small, patch-like areas of very short, sinuous, branching blood vessel impressions extending into the endocranial lamina of the skull consequent to the development of an epidural hemorrhage (Fig 1A). As the meningeal process progresses, ABVIs may be covered by appositions of newly formed bone (i.e., PAs) with a fibrous, porous, irregular, scab-like appearance (Fig 1B). In more advanced stages, the groups of tongue-like PAs with a very smooth, more mature appearance are separated by an extensive net-like aggregation of ABVIs. Bony changes resulted from hemorrhagic meningeal reactions can only be found external to the original bone surface. The inflammatory meningeal process is expressed by small, flat, plate-like appositions of newly built bone (i.e., PAs) that are oriented tangentially to the endocranial surface of the affected bone. Later, isolated

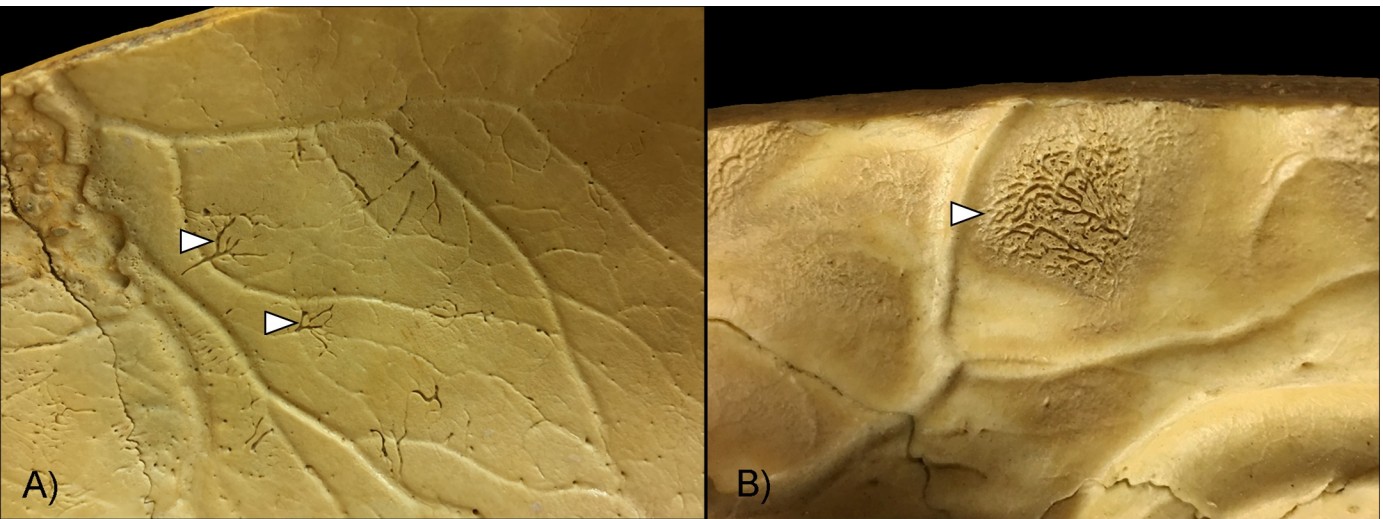

**Fig 1. Close-up of ABVIs and PAs (white arrows) on the inner surface of the skull.** A) Short, branching ABVIs on the left parietal bone (Terry No. 329, 18-year-old, male, died of pulmonary TB) and B) ABVIs accompanied by PAs on the right parietal bone (Terry No. 254, 21-year-old, male, died of pulmonary TB).

plates may be confluent as one or more layers, their contours become indistinct, and their plate-like character will be lost as a consequence of bone remodeling. In contrast to the hemorrhagic meningeal processes, the inflammatory reactions always affect the original bone surface, frequently also the deeper structures of the original bone material. Both ABVIs and PAs are generally situated in the APDIs; nevertheless, they may spread over larger areas of the skull vault in more advanced stages. Although the bony vestiges of meningeal processes may be confused by macroscopic examinations only, they can be differentiated by microscopic investigations (e.g., polarized light microscopy) into inflammatory, hemorrhagic, and mixed forms [23,25–26].

During the examination of skeletons of known cause of death from the Hamann-Todd Human Osteological Collection, Hershkovitz and his colleagues [29] observed serpentine branching surface excavations on the endocranial surface that were characterized by a maze-like appearance. These lesions were generally located in the frontal bone (predominantly around its most protruding parts), parietal bones (particularly around their most protruding part or along the superior sagittal sinus), and occipital bone (mainly along the dominant transverse sinus) and occurred predominantly in individuals who died of respiratory diseases, particularly of TB [29]. Hershkovitz and his co-workers [29] termed the above-mentioned alterations as "*serpens endocrania symmetrica*" (SES). SES may represent an advanced stage of the hemorrhagic process of the meninges: Hershkovitz and his colleagues [29] attributed it to vascular anomaly, namely to changes in the primary and secondary anastomotic arteries (traversing the outermost meningeal layer, i.e., the *dura mater*) subsequent to the development of an epidural hemorrhage. The pathological process leading to the formation of SES affects only the superficial part of the endocranial lamina of the skull with no diploic and/or ectocranial involvement [29].

Although ABVIs and PAs on the endocranial surface have been considered as paleopathological diagnostic criteria for TBM based on the findings of previous studies that were performed predominantly on osteoarchaeological series [e.g., 19,23,25–26,29,33], they are not pathognomonic features of the disease. Therefore, the utilization of ABVIs and PAs in the paleopathological practice can be questioned, especially in consideration that most of the previous studies did not assess the diagnostic value of ABVIs and PAs on skeletons of known

cause of death and/or did not include statistical data analysis. (Only Hershkovitz and his co-workers [29] conducted statistical analysis on their data but in their study, they focused only on SES that very likely represents an advanced stage of the hemorrhagic meningeal process.)

In order to expand the knowledge of ABVIs and PAs as indicators of TBM, we performed a macroscopic investigation on skeletons of known cause of death from the Terry Collection that focused on the macromorphological characteristics and frequencies of ABVIs and PAs, as well as of their co-occurrence with each other and with other probable TB-related lesions, and was completed by subsequent statistical analysis of data.

The objectives of our paper are:

1. To macroscopically evaluate the selected skeletons from the Terry Collection for the presence of ABVIs and PAs;

2. To compare the frequencies of ABVIs and PAs between individuals recorded to have died of TB versus those identified to have died of causes other than TB;

3. To macromorphologically characterize ABVIs and PAs regarding the localization, extent, and number of lesions on the affected cranial bone(s); and

4. To evaluate the diagnostic value of ABVIs and PAs.

## Materials and methods

### Materials

The Robert J. Terry Anatomical Skeletal Collection is currently curated in the Department of Anthropology at the National Museum of Natural History (Smithsonian Institution, Washington, DC, USA). It consists of 1,728 human skeletons, mostly coming from the pre-antibiotic era. For each individual, there are documentary forms providing various biographical information (e.g., age at death, sex, and cause of death) and basic anthropometric and anthroposcopic data. Furthermore, hair samples, plaster death masks, and cadaver photographs are available for approximately one half of the collection. The Terry Collection serves as an invaluable resource for anthropological and medical research, including defining and refining diagnostic criteria for specific infectious diseases, such as in this case–TB [37].

As part of a comprehensive research project [38], macromorphological characteristics, frequencies, and co-occurrences of different types of pathological bony changes probably related to TB were evaluated on all individuals (N = 302) recorded to have died of different types of TB (e.g., pulmonary TB, miliary TB, peritoneal TB, and skeletal TB). The same observations were made on a control (non-TB (NTB)) group consisting of randomly selected individuals (N = 302) from the Terry Collection, identified to have died of causes other than TB (e.g., other infectious diseases, cardiovascular problems, cancer, and external causes, such as car accident, suicide or homicide). From these 604 skeletons surveyed in the Terry Collection, 177 were excluded from the examination considering ABVIs and PAs: the skullcap was missing in two cases, the skull was not sectioned in a further 173 cases, and age at death was uncertain in two additional cases. The remaining sample consisted of 427 skeletons. The seven late adolescent (16–19 years old; three males and four females) and 420 adult (≥20 years old; 272 males and 148 females) individuals with skulls sectioned in the transverse plane (and occasionally also in the mid-sagittal plane) were divided into two main groups on the basis of their causes of death:

- TB group, consisting of 234 individuals (169 males and 65 females) recorded to have died of TB, with age at death ranging from 16 to 81 years (S1 Table) [34]; and

- Control (NTB) group, composed of 193 individuals (106 males and 87 females) identified to have died of causes other than TB, with age at death ranging from 20 to 90 years (S2 Table) [34].

## Ethics Statement

Specimen numbers: T4R, T12R, T13R, T19R, T23R, T25, T25R, T30R, T31R, T35R, T39, T44R, T46R, T47R, T54, T58R, T62RR, T64R, T69, T76R, T79R, T84, T87R, T89R, T90, T91R, T95, T95R, T103R, T104RR, T105R, T112R, T114, T124R, T127R, T128, T129, T130, T132R, T134, T135R, T138, T139, T140RR, T141R, T142R, T145R, T146R, T149R, T158R, T167, T177R, T178R, T179R, T182, T194, T197R, T199, T200, T204, T205, T207, T218, T220, T221, T222, T227, T230, T231, T232R, T235, T237, T243R, T247R, T248R, T249R, T250, T251, T254, T255, T259, T264, T265, T267, T268, T269, T270, T272, T279, T280, T282, T283R, T284, T285, T293R, T296R, T298, T304, T306, T306R, T309, T314, T317, T318, T328R, T329, T338, T339R, T341, T344R, T347, T348R, T353, T358R, T382R, T385, T386R, T393RR, T397, T400, T402, T403, T410R, T422, T423, T424, T426R, T432, T437R, T438, T444, T445, T447, T452, T453, T458, T463, T465, T466, T468, T470, T477, T483, T490, T496, T497, T504, T506, T512, T513RR, T522, T523, T527, T528, T534, T536, T537, T541, T545, T549, T552, T555, T562, T565, T566, T568, T571, T572, T573, T575, T582, T583, T585, T586, T592, T595, T597, T602, T608, T617R, T620, T621R, T626R, T627R, T629, T636, T657R, T664, T669R, T679, T680, T686, T694, T702R, T726, T727, T728R, T739, T752, T757, T759, T761, T771, T776, T786, T789, T795, T799, T809R, T820R, T822, T823, T828, T833R, T834R, T844, T846, T863, T876, T891, T892, T895, T896RR, T897, T902, T903R, T907, T914, T915, T919, T930R, T932, T933R, T934, T936, T938, T941, T946, T948, T950, T952, T955, T957, T957R, T964, T968, T973, T975, T987, T1002, T1005, T1013, T1018, T1020, T1023, T1027, T1029R, T1030, T1031, T1033, T1034, T1036, T1043, T1045, T1046, T1047, T1048, T1050, T1057, T1058, T1060, T1066R, T1070, T1071R, T1072, T1076, T1086, T1093, T1095, T1096R, T1098, T1100RR, T1102R, T1105, T1106, T1107, T1113, T1122, T1124R, T1129, T1130R, T1132, T1133RR, T1134R, T1137R, T1138R, T1140, T1147R, T1156, T1157, T1159, T1163, T1165, T1169, T1173, T1182R, T1183, T1185, T1186, T1187, T1190, T1192, T1204R, T1205, T1210, T1215, T1219, T1222, T1224, T1226, T1228, T1229, T1230, T1232, T1236, T1243R, T1247, T1249R, T1252, T1255, T1263R, T1264, T1266R, T1267, T1271, T1275, T1277, T1278, T1282, T1285, T1287, T1291, T1299R, T1300, T1304R, T1309, T1310, T1313, T1315, T1318, T1319, T1322, T1331, T1337RR, T1342, T1343, T1346, T1347R, T1352, T1353R, T1359, T1362, T1367, T1368, T1369, T1375R, T1376, T1377, T1378, T1379, T1387, T1388, T1397, T1398, T1401, T1405, T1406, T1407, T1411R, T1416, T1417R, T1419, T1422R, T1428R, T1434R, T1435, T1439R, T1444, T1451, T1453R, T1455, T1458, T1467, T1476, T1495, T1502R, T1503, T1505R, T1507, T1514, T1519, T1521, T1531, T1533, T1534, T1536, T1539, T1543, T1544, T1549, T1551, T1552, T1553, T1554, T1555, T1562, T1567, T1568, T1572, T1576, T1581, T1592, T1599, T1604, T1614, T1627, and T1629.

All specimens evaluated in the described study are housed in the Department of Anthropology, National Museum of Natural History, Smithsonian Institution, in Washington, District of Columbia, United States of America. Access to these specimens is granted by the Department of Anthropology.

No permits were required for the described study, complying with all relevant regulations.

## Methods

The endocranial surface of the 427 selected skulls was macroscopically surveyed for the presence of ABVIs and PAs. To reduce the risk of being biased, the study personnel had no

information on the cause of death of the examined individuals during the macromorphological evaluation of the 427 selected skulls. A lamp was always positioned at a distance of a few centimeters from the bone surface, since the examined bony changes can have a very subtle appearance; and therefore, it requires vigilant observation to detect their presence. For each selected individual, detailed written and pictorial descriptions of all observed ABVIs and PAs were made on a data collection sheet prepared for the current research project. The affected cranial bone(s) (considering the left and right greater wings of the sphenoid bone as two separate bones); the number of detected lesions in the affected cranial bone(s) (unifocal or multifocal); and the extent of the endocranial surface area the observed lesion(s) covered (x) in the affected cranial bone(s) (4-level scale: 1) $x < 25\%$, 2) $25\% \leq x < 50\%$, 3) $50\% \leq x < 75\%$, and 4) $75\% \leq x$) were also recorded.

Besides ABVIs and PAs, the presence of other bony lesions that can be related to different manifestations of TB (e.g., pulmonary TB, skeletal TB, and TBM) were also registered. These alterations are the following:

- Bony changes indicative of skeletal TB: osteolytic or erosive vertebral lesions, collapse or fusion of the vertebral bodies, vertebral hypervascularization, cortical remodeling and reactive new bone formation on the vertebral surfaces, signs of osteomyelitis of the extra-spinal bones, and destruction, subluxation or dislocation of the intervertebral or extra-spinal joints [e.g., 19,21,30,39–44];

- Bony lesions suggestive of pulmonary TB and/or TB pleurisy: periosteal new bone formations on the visceral surface of ribs, erosive costal changes, and signs of diffuse, symmetrical periostitis on the diaphysis of short and long tubular bones [e.g., 19–20,29,45–52]; and

- Endocranial alterations likely associated with tuberculous meningitis other than ABVIs and PAs: APDIs and GIs [e.g., 19,23–26,31,53].

After the detailed macromorphological evaluation of the 427 selected skeletons, all information collected was entered into a spreadsheet in Microsoft Office Excel 2016, and subsequent statistical analysis of data was performed. Absolute and percentage frequencies of ABVIs, PAs, and their co-occurrence with each other and with other probable TB-related lesions were calculated in both the TB group and NTB group. Moreover, $\chi^2$ testing of the data to determine the significance of differences (if any) in frequencies of ABVIs and PAs between the two groups was undertaken using the MedCalc statistical software package.

## Results

### Abnormal blood vessel impressions

During the macroscopic investigation, ABVIs were detected in 14.52% (62/427) of the skeletons examined–in 21.37% (50/234) of the TB group (S1 Table) and in 6.22% (12/193) of the NTB group (S2 Table). The $\chi^2$ testing of the frequencies of ABVIs in individuals with TB as the cause of death and individuals with NTB causes of death revealed a statistically extremely significant difference between the two groups ($\chi^2 = 18.357$, df = 1, P<0.0001).

Of the 50 individuals with ABVIs in the TB group, 42 were identified to have died of pulmonary TB (S1 Table). Five additional individuals died of other types of tuberculosis, such as peritoneal TB (three cases), TBM (one case), and miliary TB (one case); whereas in the remaining three cases, the type of tuberculosis as the cause of death was not specified on the morgue record and/or death certificate (S1 Table). In the NTB group, the most frequently registered NTB causes of death were cardiovascular problems (six cases), followed by respiratory diseases (three cases), syphilis (two cases), and different types of cancer (two cases) among individuals

exhibiting ABVIs on the inner surface of the skull–in several cases, more than one of these conditions were recorded as the cause of death (S2 Table).

Concerning the localization of ABVIs, the frontal and the left and right parietal bones (particularly their most protruding portions and/or their parts along the superior sagittal sinus (Figs 2B, 3A and 3B)) represented the most common sites of involvement in both the TB group and NTB group (S3A and S3B Table). Occasionally, the involvement of the occipital bone (generally along the superior sagittal and/or transverse sinuses (Figs 2A, 3C and 3D)) was also registered among individuals with TB as the cause of death (S3A Table). With the exception of two cases showing ABVIs on the squamous part of the right temporal bone, and two further cases on the left and right greater wings of the sphenoid bone in the TB group (S3A Table), the left and right temporal bones, and the left and right greater wings of the sphenoid bone were not affected by these lesions neither in the TB group nor in the NTB group (Fig 2A, S3A and S3B Table). The number of cranial bones concurrently involved by ABVIs (left and right sphenoid greater wings as two separate bones) varied from one to six in the TB group (Fig 4A) and from one to four in the NTB group (Fig 4B). In nearly two-thirds (32/50, 64.00%) of individuals with TB as the cause of death and ABVIs, at least three cranial bones were simultaneously affected (Fig 4A); whereas in three-fourths (9/12, 75.00%) of individuals

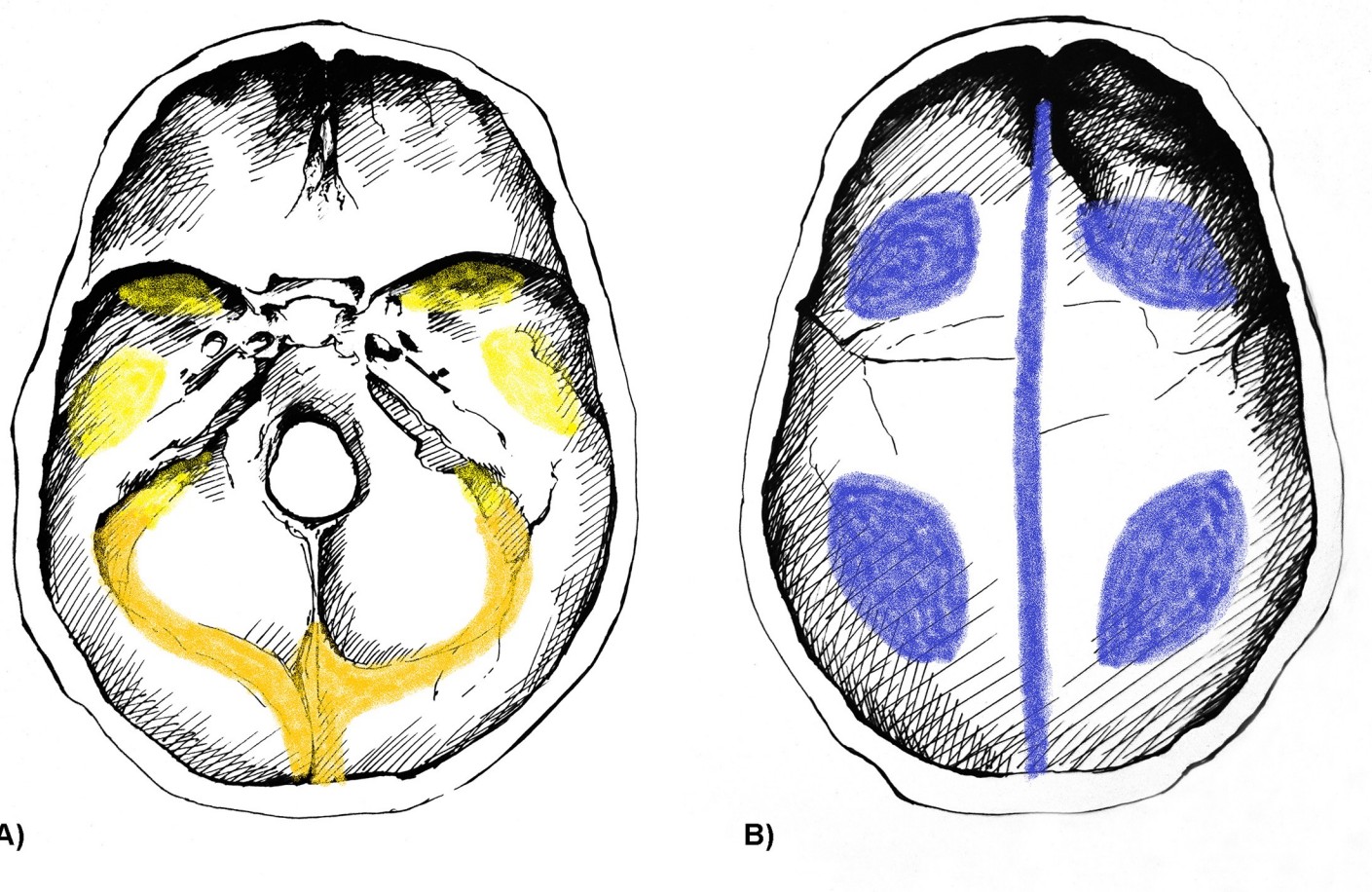

**A)**    **B)**

**Fig 2.** Typical localizations of ABVIs on the inner surface of the A) skull base and B) skullcap. Blue: most commonly affected areas, orange: commonly affected areas, and yellow: less commonly affected areas (drawings by Luca Kis).

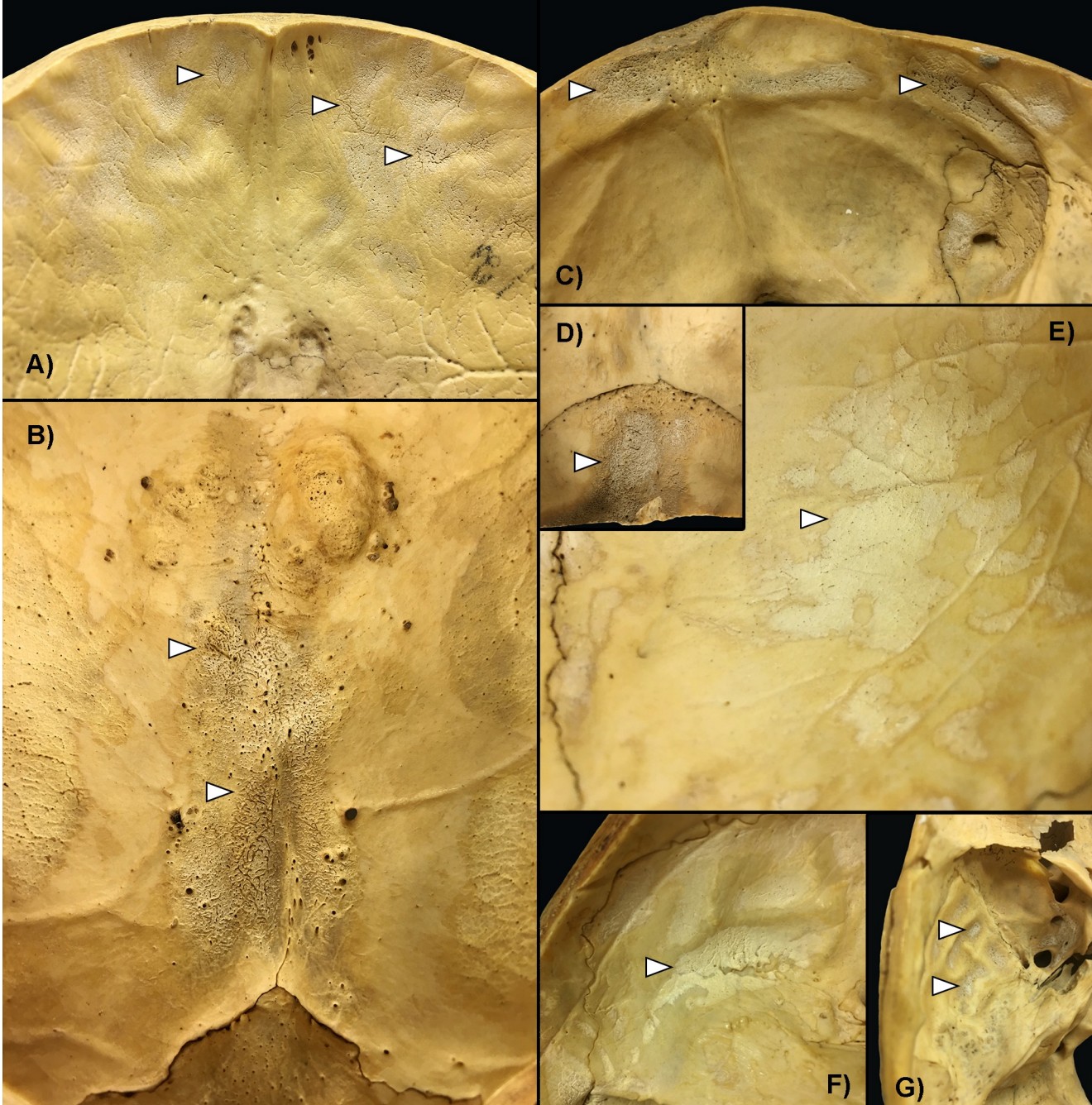

**Fig 3. Typical localizations of ABVIs and PAs (white arrows) on the inner surface of the skull.** A) The squamous part of the frontal bone (Terry No. 304, 20-year-old, female, died of pulmonary TB); B) The area along the superior sagittal sinus on the parietal bones (Terry No. 1222, 28-year-old, female, died of pulmonary TB); C) The squamous part of the occipital bone (lower part) (Terry No. 1322, 34-year-old, male, died of pulmonary TB); D) The squamous part of the occipital bone (upper part) (Terry No. 1322, 34-year-old, male, died of pulmonary TB); E) The most protruding portion of the parietal bone (Terry No. 987, 23-year-old, male, died of pulmonary TB); F) The squamous part of the temporal bone (Terry No. 987, 23-year-old, male, died of pulmonary TB); and G) The squamous part of the temporal bone and the greater wing of the sphenoid bone (Terry No. 1322, 34-year-old, male, died of pulmonary TB).

with NTB causes of death and ABVIs, less than three cranial bones were concomitantly involved (Fig 4B).

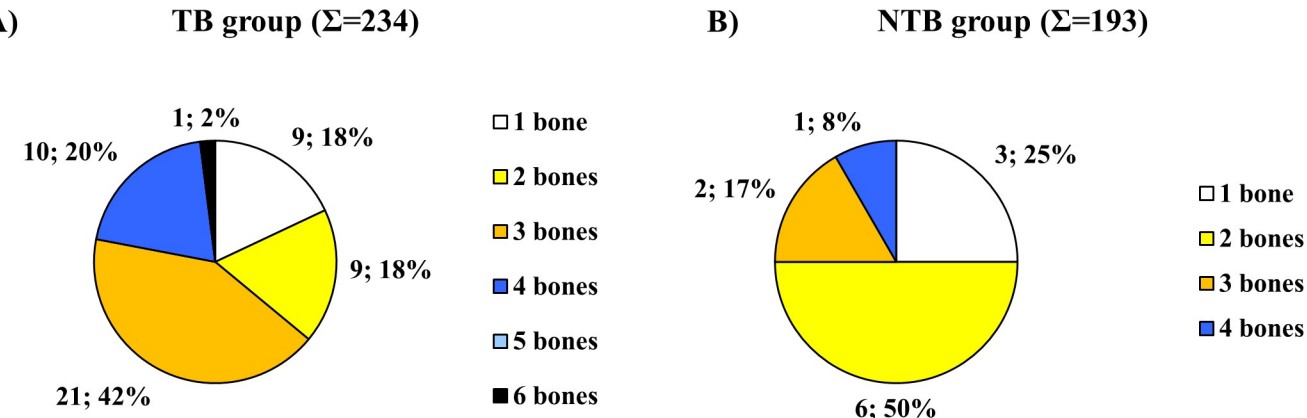

**Fig 4.** Distribution of individuals affected by ABVIs in the A) TB group (Σ = 50) and B) NTB group (Σ = 12) by number of simultaneously involved cranial bones (considering the left and right greater wings of the sphenoid bone as two separate bones).

Regarding the number of presented lesions, ABVIs were almost exclusively detected as multifocal bony changes on the inner surface of the frontal, parietal, and occipital bones in both the TB group and NTB group (S3A and S3B Table). Nonetheless, unifocal ABVIs occurred in the left and right parietal, right temporal, and/or occipital bones in a few cases in both individuals identified to have died of TB and individuals recorded to have died of causes other than TB (S3A and S3B Table). With respect to the extent of the registered lesions, the majority of the observed ABVIs covered less than one-half of the endocranial surfaces in all cranial bones examined in both the TB group and NTB group (S3A and S3B Table). Nevertheless, in the TB group, the extent of ABVIs noted in the frontal and the left and right parietal bones exceeded one-half of the inner surfaces quite often: in 30.95% (13/42), 18.42% (7/38), and 19.44% (7/36) of cases, respectively (S3A Table).

In 49 out of the 50 individuals (98.00%) with ABVIs in the TB group, ABVIs concomitantly occurred with other probable TBM-related endocranial alterations (APDIs: 36 cases, PAs: 24 cases, and GIs: 20 cases (Fig 5A and 5B and S5 Table)) and/or possible TB-associated non-endocranial bony changes (periosteal new bone formations on the visceral surface of ribs: 33 cases, vertebral hypervascularization: 33 cases, vertebral lytic lesions and/or arthritis: six cases, signs of extra-spinal osteomyelitis: two cases, signs of extra-spinal arthritis: three cases, signs of

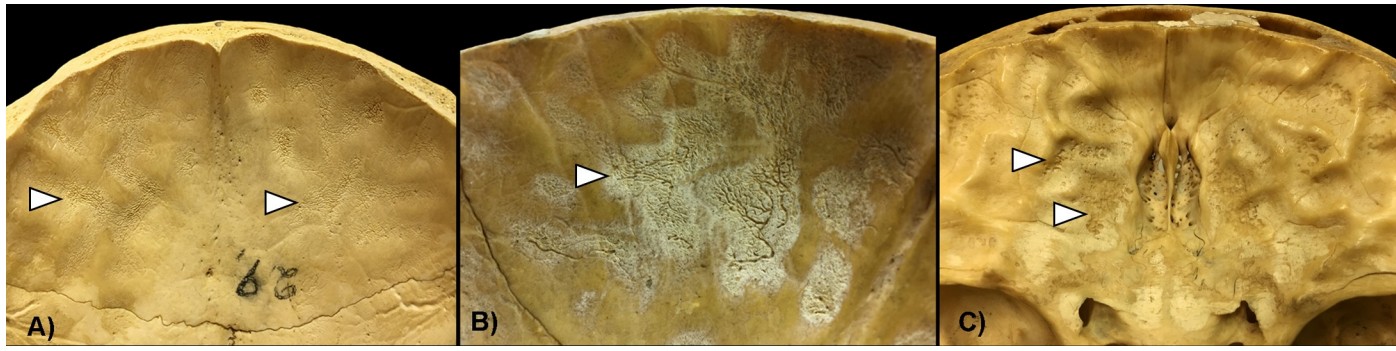

**Fig 5. Co-occurrence of ABVIs and PAs with each other and with other probable TBM-related bony changes (white arrows).** A) ABVIs and PAs in the abnormally pronounced digital impressions on the squamous part of the frontal bone (Terry No. 254, 21-year-old, male, died of pulmonary TB); B) ABVIs and PAs on the squamous part of the frontal bone (Terry No. 306, 18-year-old, female, died of pulmonary TB); and C) PAs accompanied by granular impressions on the orbital part of the frontal bone (Terry No. 1159, 26-year-old, male, died of pulmonary TB).

**A)      TB group (Σ=234)**

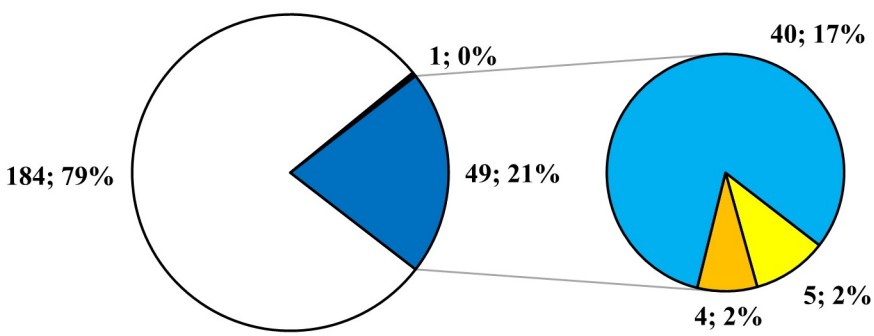

□ **Without ABVIs**

■ **With ABVIs and without other probable TB-related endocranial or non-endocranial lesion(s)**

□ **With ABVIs and with other probable TB-related endocranial lesion(s)**

□ **With ABVIs and with other probable TB-related non-endocranial lesion(s)**

■ **With ABVIs and with other probable TB-related endocranial and non-endocranial lesion(s)**

**B)      NTB group (Σ=193)**

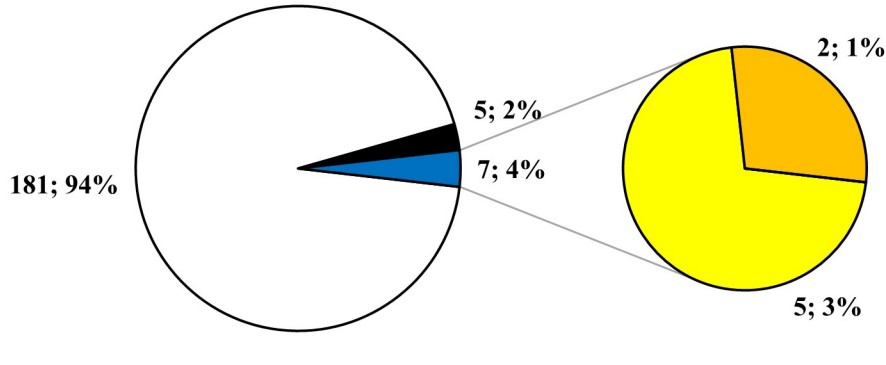

□ **Without ABVIs**

■ **With ABVIs and without other probable TB-related endocranial or non-endocranial lesion(s)**

□ **With ABVIs and with other probable TB-related endocranial lesion(s)**

□ **With ABVIs and with other probable TB-related non-endocranial lesion(s)**

■ **With ABVIs and with other probable TB-related endocranial and non-endocranial lesion(s)**

**Fig 6.** Distribution of individuals exhibiting ABVIs in the A) TB group (Σ = 50) and B) NTB group (Σ = 12) by number of presented probable TBM-related endocranial lesion types other than ABVIs and/or of possible TB-associated non-endocranial lesion types.

hypertrophic pulmonary osteopathy: 11 cases, and signs of cold abscesses: seven cases (S7 Table)) (Fig 6A). In seven out of the 12 individuals (58.33%) with ABVIs in the NTB group, probable TBM-related endocranial alterations other than ABVIs (APDIs: three cases and PAs: two cases (S9 Table)) and/or possible TB-associated non-endocranial bony changes (periosteal new bone formations on the visceral surface of ribs: one case, vertebral lytic lesions and/or arthritis: one case, signs of extra-spinal osteomyelitis: one case, signs of hypertrophic pulmonary osteopathy: one case, and signs of cold abscesses: one case (S11 Table)) were concurrently registered (Fig 6B).

## Periosteal appositions

From a total of 427 skeletons evaluated in the Terry Collection, 67 (15.69%) exhibited PAs on the inner skull surface: 47 (20.09%) of 234 individuals recorded to have died of TB (S1 Table) and 20 (10.36%) of 193 individuals identified to have died of causes other than TB (S2 Table).

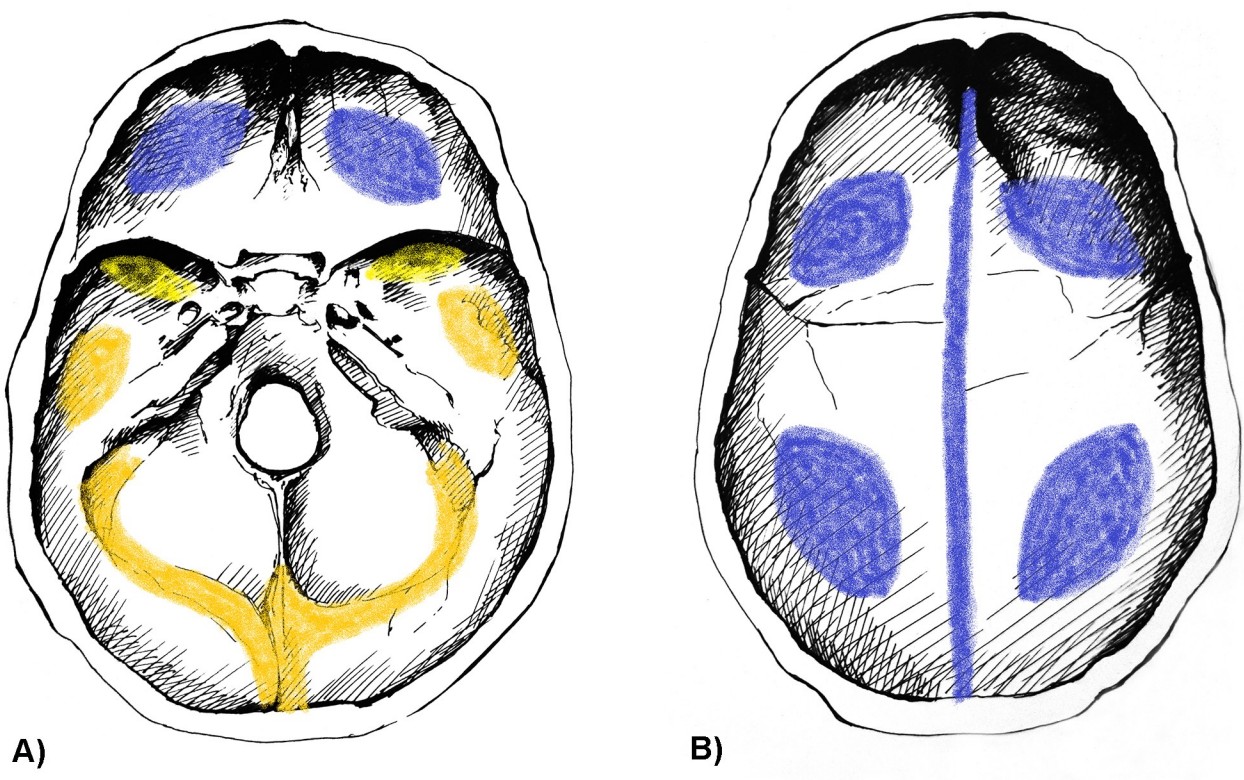

**Fig 7.** Typical localizations of PAs on the inner surface of the A) skull base and B) skullcap. Blue: most commonly affected areas, orange: commonly affected areas, and yellow: less commonly affected areas (drawings by Luca Kis).

The $\chi^2$ comparison of the frequencies of PAs revealed a statistically significant difference between the two groups ($\chi^2$ = 6.841, df = 1, P = 0.0089).

Among individuals with PAs in the TB group, the type of tuberculosis as the cause of death was not specified on the morgue record and/or death certificate in six cases (S1 Table). In the remaining 41 cases–with the exception of two individuals who died of TBM and an individual who died of skeletal TB–pulmonary TB (38 cases) was registered as the cause of death (S1 Table). Among individuals with PAs in the NTB group, cardiovascular problems (14 cases), pneumonia (three cases), cancer (one case), syphilis (one case), and appendicitis (one case) were recorded as NTB causes of death (S2 Table).

Regarding the localization of PAs on the inner surface of the skull, the frontal bone (predominantly its most protruding portions and/or orbital part (Figs 3A, 7A and 7B)) and the left and right parietal bones (particularly their most protruding portions and/or their parts along the superior sagittal sinus (Figs 3B, 3E and 7B)) represented the most common sites of involvement in both the TB group and NTB group (S4A and S4B Table). Less often, the occipital bone (generally its squamous part (Figs 3C, 3D and 7A)), the left and right temporal bones (predominantly their squamous parts (Figs 3F, 3G and 7A)), and more rarely the left and right greater wings of the sphenoid bone (Figs 3G and 7A) were also affected. The number of cranial bones concurrently involved by PAs (left and right sphenoid greater wings as two separate bones) varied from one to eight in both groups (Fig 8). With 53.19% (25/47) of individuals with TB as the cause of death and PAs, at least four cranial bones were simultaneously affected (Fig 8A). In contrast, in nearly two-thirds (13/20, 65.00%) of individuals with NTB causes of death and PAs, less than four cranial bones were concomitantly involved (Fig 8B).

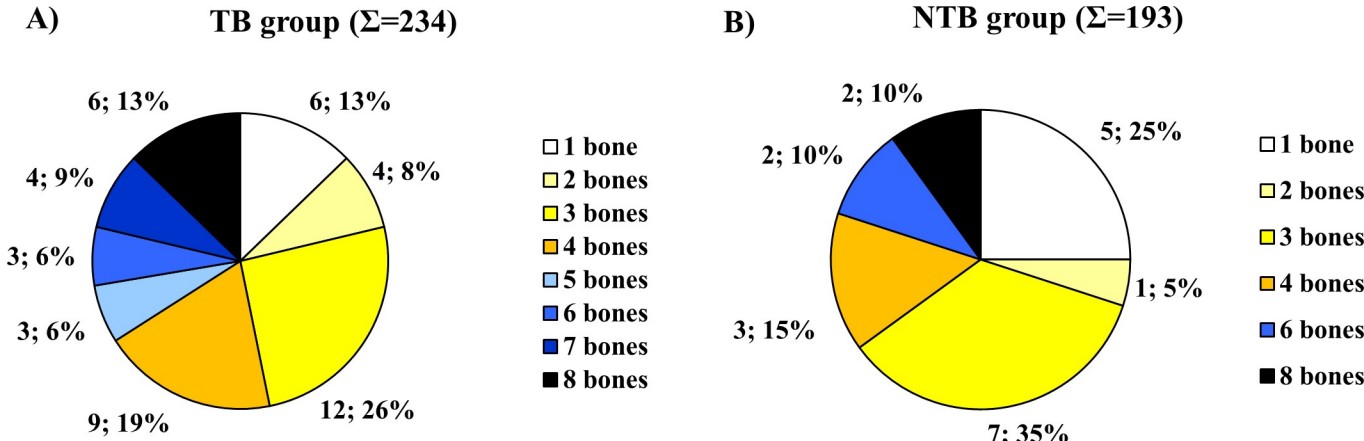

**Fig 8.** Distribution of individuals affected by PAs in the A) TB group (Σ = 47) and B) NTB group (Σ = 20) by number of simultaneously involved cranial bones (considering the left and right greater wings of the sphenoid bone as two separate bones).

With respect to the number of presented lesions in the TB group, the vast majority of PAs occurred as multifocal alterations on the inner surface of the skull in all cranial bones examined, except for the left and right greater wings of the sphenoid bone, where the ratio of multifocal and unifocal PAs was about 3:2 (S4A Table). Among individuals recorded to have died of causes other than TB, almost exclusively multifocal PAs were registered in all cranial bones evaluated (S4B Table). Concerning the extent of the detected lesions, the majority of PAs observed in the TB group covered less than one-half of the endocranial surfaces in all cranial bones examined (S4A Table). Nonetheless, the extent of PAs noted around the most protruding portions of the frontal and the left and right parietal bones, as well as in the squamous part of the left temporal bone, exceeded one-half of the inner surfaces quite often: in 20.59% (7/34), 23.68% (9/38), 20.00% (7/35), and 22.22% (4/18) of cases, respectively (S4A Table). Among individuals with NTB causes of death, only four PAs detected around the most protruding portions of the frontal and the left and right parietal bones covered more than one-half of the endocranial surfaces (S4B Table).

In all the 47 individuals (100.00%) with PAs in the TB group, PAs simultaneously occurred with other probable TBM-related endocranial alterations (APDIs: 43 cases, ABVIs: 24 cases, and GIs: 12 cases (Fig 5 and S6 Table)) and/or possible TB-associated non-endocranial bony changes (periosteal new bone formations on the visceral surface of ribs: 30 cases; vertebral hypervascularization: 29 cases; vertebral lytic lesions and/or arthritis: four cases; signs of extra-spinal osteomyelitis: one case; signs of extra-spinal arthritis: two cases; signs of hypertrophic pulmonary osteopathy: seven cases; and signs of cold abscesses: two cases (S8 Table)) (Fig 9A). In ten out of the 20 individuals (50.00%) with PAs in the NTB group, probable TBM-related endocranial alterations other than PAs (APDIs: seven cases, ABVIs: two cases, and GIs: one case (S10 Table)) and/or possible TB-associated non-endocranial bony changes (periosteal new bone formations on the visceral surface of ribs: one case and vertebral hypervascularization: one case (S12 Table)) were concomitantly recorded (Fig 9B).

## Discussion and conclusions

In the paleopathological literature [e.g., 19,23–26,29–31], since the late 20th century only a few endocranial alteration types, including ABVIs and PAs, were described as probable signs of TBM. TB involvement of the leptomeninges (i.e., the *pia* and *arachnoid mater*), also known as

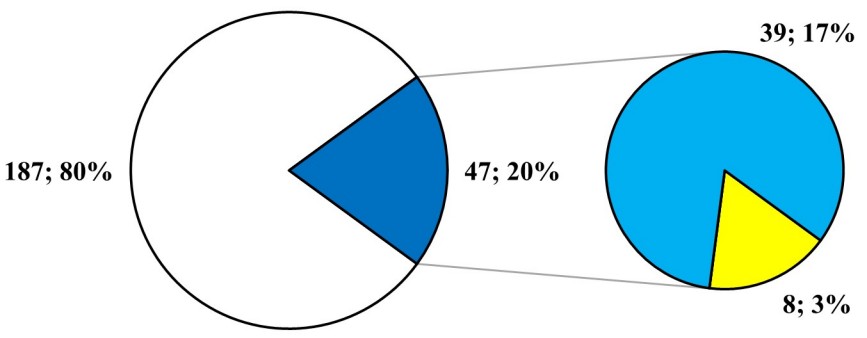

**A) TB group (Σ=234)**

□ Without PAs

■ With PAs and without other probable TB-related endocranial or non-endocranial lesion(s)

□ With PAs and with other probable TB-related endocranial lesion(s)

■ With PAs and with other probable TB-related non-endocranial lesion(s)

□ With PAs and with other probable TB-related endocranial and non-endocranial lesion(s)

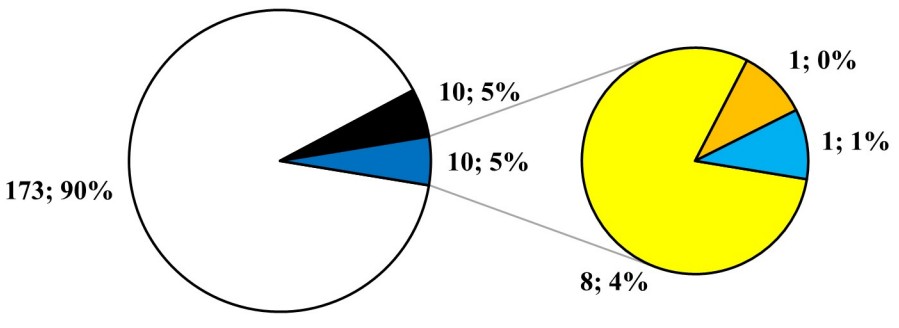

**B) NTB group (Σ=193)**

□ Without PAs

■ With PAs and without other probable TB-related endocranial or non-endocranial lesion(s)

□ With PAs and with other probable TB-related endocranial lesion(s)

■ With PAs and with other probable TB-related non-endocranial lesion(s)

□ With PAs and with other probable TB-related endocranial and non-endocranial lesion(s)

**Fig 9.** Distribution of individuals exhibiting PAs in the A) TB group (Σ = 47) and B) NTB group (Σ = 20) by number of presented probable TBM-related endocranial lesion types other than PAs and/or of possible TB-associated non-endocranial lesion types.

*leptomeningitis tuberculosa*, is the most common form of central nervous system (CNS) TB, accounting for about 70–80% of the cases [54–60]. TBM usually develops subsequent to the rupture of one or more meningeal, subpial, and/or subependymal caseating tubercles into the subarachnoid space or into the ventricular system, both occupied by the cerebrospinal fluid (CSF) [54–55,57–69]. The release of sufficient numbers of TB bacteria into the CSF triggers the onset of diffuse granulomatous inflammation of the leptomeninges, with strong predilection for the basal areas of the brain [54–55,62,65–68,70]. Nonetheless, not only the *pia* and *arachnoid mater* but additionally, the outermost meningeal layer (i.e., the *dura mater*) can be affected by the disease [71]. Besides the small tubercles primarily formed in the leptomeninges and later also in the *dura mater*, characteristic pathological features of TBM include enhancing basal meningeal exudate, progressive hydrocephalus, and vasculitis of the blood vessels adjacent to or traversing the exudate [57,59–60,62,64,67,69–71]. In TBM, not only the intracranial arteries and veins but also the meningeal vessels and dural venous sinuses can become inflamed, occasionally resulting in hemorrhages [72–73]. The inflammatory and/or hemorrhagic processes of the meninges in TBM can generate the formation of ABVIs and PAs on the

inner surface of the skull [e.g., 19,23–26,35–36]. However, these endocranial alteration types cannot be considered as specific vestiges of TBM, since other pathological conditions (e.g., trauma and scurvy) can also lead to their development. Consequently, the diagnostic value of ABVIs and PAs in the paleopathological identification of TBM can be questioned.

To assess the diagnostic value of ABVIs and PAs, we performed a macroscopic investigation on identified skeletons from the Terry Collection that focused on the macromorphological characteristics and frequency of ABVIs, PAs, as well as of their co-occurrence with each other and with other probable TB-related lesions. This investigation was completed by subsequent statistical analysis of data.

During the evaluation of the 427 selected skeletons with sectioned skulls from the Terry Collection, we found that about one-seventh of the surveyed individuals exhibited ABVIs on the endocranial surface and ABVIs were registered in both the TB group and NTB group. Based on our findings, ABVIs do have a diagnostic value in the paleopathological identification of TBM in osteoarchaeological material from the pre-antibiotic era. ABVIs were about three and a half times more common in individuals with TB as the cause of death than in individuals with NTB causes of death. The $\chi^2$ comparison of the frequencies of ABVIs revealed a statistically extremely significant difference between the TB group and NTB group–this indicates a positive association between ABVIs and TBM. In addition, in the vast majority of the 50 individuals with ABVIs in the TB group, ABVIs concomitantly occurred with other probable TBM-related endocranial alterations (Figs 5A, 5B and 6A and S5 Table) and/or possible TB-associated non-endocranial bony changes (Fig 6A and S7 Table). This further supports the tuberculous origin of ABVIs observed in the TB group.

Our results support those of previous studies [e.g., 23–26] regarding the specificity of ABVIs for TBM. More than 6% of the skeletons composing the NTB group also exhibited ABVIs on the inner surface of the skull. This suggests that ABVIs cannot be considered as pathognomonic features of TBM. In more than one-half of the 12 individuals with ABVIs in the NTB group, probable TBM-related endocranial alterations other than ABVIs (Fig 6B and S9 Table) and/or likely TB-associated non-endocranial bony changes (Fig 6B and S11 Table) were also noted. It must be mentioned that even if the recorded cause of death of individuals surveyed in the Terry Collection may not have been TB, individuals could still have suffered from the disease but their death was attributed to another medical condition [46,50]. Therefore, it cannot be excluded that in the above-mentioned seven individuals with ABVIs in the NTB group, the observed endocranial and non-endocranial bony changes might result from TB. However, in the other five individuals with ABVIs in the NTB group, where no signs of probable TBM-related endocranial alterations other than ABVIs or of possible TB-associated non-endocranial bony changes were detected, the NTB origin (e.g., CNS infections other than TBM, brain tumors, and hemorrhages) of the noted lesions is much more likely.

Of the 427 selected skeletons with sectioned skulls from the Terry Collection, about one-sixth exhibited PAs on the endocranial surface and PAs were registered in both the TB group and NTB group. Our results indicate that, similar to ABVIs, PAs do have a diagnostic value in the paleopathological identification of TBM in ancient human bone remains, since they were about twice more common in individuals with TB as the cause of death than in individuals with NTB causes of death. The $\chi^2$ comparison of the frequencies of PAs revealed a statistically significant difference between the TB group and NTB group that suggests a positive association between PAs and TBM. Furthermore, in all the 47 individuals with PAs in the TB group, PAs simultaneously occurred with other probable TBM-related endocranial alterations (Figs 5, 9A and S6 Table) and/or possible TB-associated non-endocranial bony changes (Fig 9A, S8 Table). This further supports the tuberculous origin of PAs observed in the TB group.

Our findings concur with those of previous studies [e.g., 23–26,35] concerning the specificity of PAs for TBM. PAs were detected in more than 10% of the skeletons composing the NTB group. This suggests that PAs cannot be considered as specific vestiges of TBM. In one-half of the 20 individuals with PAs in the NTB group, probable TBM-related endocranial alterations other than PAs (Fig 9B, S10 Table) and/or possible TB-associated non-endocranial bony changes (Fig 9B and S12 Table) were concomitantly recorded. Since the disease registered as the cause of death on the morgue record and/or death certificate may not have been the only medical condition present in the individuals surveyed in the Terry Collection, individuals identified to have died of causes other than TB could still have suffered from TB at death [46,50]. Thus, in the aforementioned ten individuals with PAs in the NTB group, the tuberculous origin of the recorded endocranial and non-endocranial lesions cannot be excluded. It is also possible that in these ten cases (similar to the other ten individuals with PAs in the NTB group, where no signs of probable TBM-related endocranial alterations other than PAs or of likely TB-associated non-endocranial bony changes were observed) medical conditions other than TB (e.g., bacterial meningitis, trauma, and scurvy) resulted in the development of the detected lesions.

It should be mentioned that our research project has some limitations. One of its major limitations is the absence of children in the examined skeletal material, since, according to the modern medical literature [61–62,66,74–75], children under the age of five years represent the most vulnerable group affected by TBM. Unfortunately, the composition of the Terry Collection [37] did not make it possible to evaluate child skeletons (the youngest individual of the Terry Collection died at the age of 14 years). Nevertheless, in the future, it would be very useful to examine identified child skeletons to determine whether or not the frequency and macromorphological characteristics of ABVIs and PAs are similar to that of observed in adults in the Terry Collection. The chance of the formation and extension of an epidural hemorrhage is dependent on the strength of the adherence of the *dura mater* to different areas of the endocranial lamina of the skull that changes with age [76–78]. This may have an impact on the frequency and macromorphological characteristics (e.g., location, extent, and number) of ABVIs and PAs in different age groups. Furthermore, our findings regarding the description of the macromorphological characteristics of ABVIs and PAs could be further improved by applying not only macromorphological methods but also imaging (e.g., micro-CT) and microscopy (e.g., polarized light microscopy) techniques in the investigation of osteoarchaeological and documented skeletal collections. Some studies on osteoarchaeological series [e.g., 79–82], using not only macromorphological but also microscopy methods (e.g., scanning electron microscopy and thin-ground section microscopy [cf., 83]), have already revealed that the vestiges of meningeal processes are much more common in children than in adults and the appearance of these endocranial alterations is much more diverse than based only on macromorphology.

The other major limitation of our research project is the low number of individuals recorded to have died of TBM in the TB group. It is not surprising if we consider that TBM occurs in less than 1% of all cases with active TB [74]. Nonetheless, some autopsy studies revealed that a large number of individuals who died of pulmonary TB without developing neurological signs and symptoms exhibited tubercles in the CNS. The results of our recently published study [34], also conducted on the Terry Collection, fit in with those of the aforementioned autopsy studies. Although the vast majority of the individuals in our TB group were identified to have died of pulmonary TB (S1 Table), about one-third of them exhibited GIs that can be considered as pathognomonic features of TBM. These indicate that involvement of the CNS in pulmonary TB is quite common [68]. Moreover, some recent studies showed that about three-fourths of the patients with CNS TB had pulmonary TB 6–12 months prior to the

onset of neurological signs and symptoms [58]. Therefore, the presence of GIs, ABVIs, and/or PAs on the inner surface of the skull in individuals with pulmonary TB as the cause of death from the Terry Collection suggests that these individuals could already have CNS involvement without developing neurological signs and symptoms prior to death.

In summary, although further investigations on human skeletons from documented collections other than the Terry Collection are necessary to confirm the trends noted in our study, our results constitute evidence that there is a positive association between ABVs and TB, as well as between PAs and TB. Thus, ABVIs and PAs can be used as diagnostic criteria for TBM in the paleopathological practice. Although they cannot be considered as pathognomonic features of the disease, paleopathologists could still use them to diagnose TB in human osteoarchaeological series from the pre-antibiotic era, especially when they simultaneously occur with other endocranial and/or non-endocranial bony changes probably related to TB. Similar to other paleopathological diagnostic criteria for TB that are not specific to the disease (e.g., periosteal new bone formations on the visceral surface of ribs [e.g., 20,45–46,50–51] and vertebral hypervascularization [e.g., 19,21,39,41,43]), utilization of ABVIs and PAs–with suitable circumspection–provides paleopathologists with a stronger basis for identifying TB and consequently, with a more sensitive means of assessing TB frequency in past human populations. In addition, refinement of macromorphological diagnostic criteria for TB and their application in the paleopathological practice may open new perspectives in the evolutionary research of TB, since TB cases identified in human osteoarchaeological series from prehistoric and historic times can serve as an invaluable source for ancient DNA of members of the *Mycobacterium tuberculosis* complex (i.e., bacterial species causing TB in human and animal hosts–MTBC). By gaining a better understanding of the origin and evolutionary history of the MTBC, as well as of the co-evolution of its members with the human host, there is a great opportunity before us to eliminate or at least control TB in the foreseeable future.

## Supporting information

**S1 Table. Basic biographic data of individuals in the TB group (N = 234).** (MR = morgue record; DC1 = death certificate primary; DC2 = death certificate secondary; DC3 = death certificate tertiary; c. = circa; F = female; M = male; TB = tuberculosis; ABVIs = abnormal blood vessel impressions; PAs = periosteal appositions; + = exhibiting ABVIs/PAs;– = not exhibiting ABVIs/PAs).
(PDF)

**S2 Table. Basic biographic data of individuals in the NTB group (N = 193).** (MR = morgue record; DC1 = death certificate primary; DC2 = death certificate secondary; DC3 = death certificate tertiary; c. = circa; F = female; M = male; TB = tuberculosis; NTB = non-tuberculous; ABVIs = abnormal blood vessel impressions; PAs = periosteal appositions; + = exhibiting ABVIs/PAs;– = not exhibiting ABVIs/PAs).
(PDF)

**S3 Table Distribution of individuals exhibiting ABVIs in the Terry Collection by affected cranial bones (considering the left and right greater wings of the sphenoid bone as two separate bones), extent, and number of lesions (TB = tuberculosis; NTB = non-tuberculous; ABVIs = abnormal blood vessel impressions; L = left; R = right) Number of individuals in the A) TB group and B) NTB group.**
(PDF)

**S4 Table Distribution of individuals exhibiting PAs in the Terry Collection by affected cranial bones (considering the left and right greater wings of the sphenoid bone as two**

separate bones), extent, and number of lesions (TB = tuberculosis; NTB = non-tuberculous; PAs = periosteal appositions; L = left; R = right) Number of individuals in the A) TB group and B) NTB group.
(PDF)

**S5 Table. Individual data of cases exhibiting ABVIs regarding other probable TBM-associated endocranial bony changes in the TB group (Σ = 50).** (TB = tuberculosis; TBM = tuberculous meningitis; ABVIs = abnormal blood vessel impressions; APDIs = abnormally pronounced digital impressions; PAs = periosteal appositions; GIs = granular impressions; + = present; − = not present).
(PDF)

**S6 Table. Individual data of cases exhibiting PAs regarding other probable TBM-associated endocranial bony changes in the TB group (Σ = 47).** (TB = tuberculosis; TBM = tuberculous meningitis; PAs = periosteal appositions; APDIs = abnormally pronounced digital impressions; ABVIs = abnormal blood vessel impressions; GIs = granular impressions; + = present; − = not present).
(PDF)

**S7 Table. Individual data of cases exhibiting ABVIs on the inner skull surface regarding possible TB-related non-endocranial bony changes in the TB group (Σ = 50).**
(TB = tuberculosis; ABVIs = abnormal blood vessel impressions; PNBFs = periosteal new bone formations; HPO = hypertrophic pulmonary osteopathy; + = present; − = not present).
(PDF)

**S8 Table. Individual data of cases exhibiting PAs on the inner skull surface regarding possible TB-related non-endocranial bony changes in the TB group (Σ = 47).**
(TB = tuberculosis; PAs = periosteal appositions; PNBFs = periosteal new bone formations; HPO = hypertrophic pulmonary osteopathy; + = present; − = not present).
(PDF)

**S9 Table. Individual data of cases exhibiting ABVIs regarding other probable TBM-associated endocranial bony changes in the NTB group (Σ = 12).** (NTB = non-tuberculous; TBM = tuberculous meningitis; ABVIs = abnormal blood vessel impressions; APDIs = abnormally pronounced digital impressions; PAs = periosteal appositions; GIs = granular impressions; + = present; − = not present).
(PDF)

**S10 Table. Individual data of cases exhibiting PAs regarding other probable TBM-associated endocranial bony changes in the NTB group (Σ = 20).** (NTB = non-tuberculous; TBM = tuberculous meningitis; PAs = periosteal appositions; APDIs = abnormally pronounced digital impressions; ABVIs = abnormal blood vessel impressions; GIs = granular impressions; + = present; − = not present).
(PDF)

**S11 Table. Individual data of cases exhibiting ABVIs on the inner skull surface regarding possible TB-related non-endocranial bony changes in the NTB group (Σ = 12).**
(NTB = non-tuberculous; ABVIs = abnormal blood vessel impressions; PNBFs = periosteal new bone formations; HPO = hypertrophic pulmonary osteopathy; + = present; − = not present).
(PDF)

**S12 Table. Individual data of cases exhibiting PAs on the inner skull surface regarding possible TB-related non-endocranial bony changes in the NTB group (Σ = 20).** (NTB = nontuberculous; PAs = periosteal appositions; PNBFs = periosteal new bone formations; HPO = hypertrophic pulmonary osteopathy; + = present; − = not present).
(PDF)

## Acknowledgments

Special thanks go to Luca Kis for the drawings in Figs 2 and 7.

## Author Contributions

**Conceptualization:** Olga Spekker.

**Data curation:** Olga Spekker, David R. Hunt.

**Formal analysis:** Olga Spekker, László Paja.

**Funding acquisition:** Olga Spekker, György Pálfi.

**Investigation:** Olga Spekker.

**Methodology:** Olga Spekker.

**Project administration:** Olga Spekker.

**Resources:** David R. Hunt.

**Supervision:** Michael Schultz, Erika Molnár, György Pálfi, David R. Hunt.

**Visualization:** Olga Spekker, László Paja, Orsolya A. Váradi.

**Writing – original draft:** Olga Spekker, David R. Hunt.

**Writing – review & editing:** Olga Spekker.

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
