## [Decision Letter · Decision Letter 0]

12 Aug 2020

PONE-D-20-18111

Tracking down the White Plague. Chapter two: The role of endocranial abnormal blood vessel impressions and periosteal appositions in the paleopathological diagnosis of tuberculous meningitis

PLOS ONE

Dear Dr. Spekker,

Thank you for submitting your manuscript to PLOS ONE. After careful consideration, we feel that it has merit but does not fully meet PLOS ONE’s publication criteria as it currently stands. Therefore, we invite you to submit a revised version of the manuscript that addresses the points raised during the review process.

The manuscript as presented is high quality and of interest to the overall readership of Plos One. There are some minor considerations included in the review that the authors should address.

We look forward to receiving your revised manuscript.

Kind regards,

JJ Cray Jr., Ph.D.

Academic Editor

PLOS ONE

Journal Requirements:

2. In your manuscript, please provide additional information regarding the specimens used in your study. Ensure that you have reported specimen numbers and complete repository information, including museum name and geographic location.

For more information on PLOS ONE's requirements for paleontology and archaeology research, see https://journals.plos.org/plosone/s/submission-guidelines#loc-paleontology-and-archaeology-research.

Reviewers' comments:

Reviewer's Responses to Questions

**Comments to the Author**

1. Is the manuscript technically sound, and do the data support the conclusions?

Reviewer #1: Yes

2. Has the statistical analysis been performed appropriately and rigorously? 

Reviewer #1: Yes

3. Have the authors made all data underlying the findings in their manuscript fully available?

Reviewer #1: Yes

4. Is the manuscript presented in an intelligible fashion and written in standard English?

Reviewer #1: Yes

5. Review Comments to the Author

Reviewer #1: This article examines the relationship between tuberculosis and two classes of endocranial lesions, abnormal blood vessel impressions and periosteal appositions. The authors use the Terry Collection, which represents something of a control sample, coding individuals by the listed cause of death, which as the authors mention, is not necessarily synonymous with how cause of death would be diagnosed and/or recorded today using 21st century medical understanding of disease and mortality. The ultimate goal, is to then to make these findings available to paleopathologists and ostearchaeologists, who are nearly always dealing with skeletal remains where the cause of death is unknown. The hope is that endocranial lesion frequency, context, and morphology can be used to help diagnose more ancient disease and/or causes of death.

I enjoyed reading the paper and found it quite interesting. Overall, I find the study well-designed, the statistical analysis simple but adequate, the text to be very well-written and concise, and the results significant and of great interest to the paleopathology field. I have no hesitation in recommending its publication in PLOS-ONE. I make a few remarks below that the authors might consider in moving towards a final publication. They are merely suggestions.

I wonder if the authors could pull out from the NTB group, specifically, those that died from accidents, and analyze them as a subgroup (in addition to the entire NTB group)? While individuals dying from accidents could still have some underlying TB infection (or NTB disease associated with meningeal infection), it might represent a better “control” group for the specific goals of this paper. On the other hand, the sample size of the NTB-accident cohort may be so small, that statistical comparison is not possible.

Are any of the individuals in the collection specifically identified as having died from meningococcal disease (Neisseria meningitidis infection), or other diseases associated with meningeal infections?

Lines 228-231: for the NTB group, could you give in parentheses (n=X) the number of individuals with ABVIs dying of cardiovascular, respiratory disease, syphilis, etc.? So we don’t have to go to the supplementary table.

It would be helpful in the figures of bones to add small arrows to draw the reader’s attention specifically to the features the authors wish to highlight in each frame.

6. PLOS authors have the option to publish the peer review history of their article (what does this mean?). If published, this will include your full peer review and any attached files.

Reviewer #1: No

---

## [Author Response · Author response to Decision Letter 0]

15 Aug 2020

Dr. JJ Cray Jr., PhD

Academic Editor

PLOS ONE

August 13, 2020

Dear Dr. JJ Cray Jr.,

I am very thankful for the reviewer’s insightful and constructive comments regarding our manuscript entitled “Tracking down the White Plague. Chapter two: The role of endocranial abnormal blood vessel impressions and periosteal appositions in the paleopathological diagnosis of tuberculous meningitis” that was submitted to PLOS ONE (manuscript ID: PONE-D-20-18111). I am sure that the reviewer helped us to improve the quality of our manuscript. The main text has been modified following the reviewer’s suggestions, and the revised version of our manuscript has been uploaded to the submission site of PLOS ONE.

Responses to the suggestions:

1) Reviewer 1 mentioned that individuals from the NTB group who died of accidents could be analysed as a subgroup, and they might represent a better “control” group for the specific goals of our paper. Following Reviewer 1’s request, we pulled out the aforementioned individuals from the NTB group. They are the following:

1) Terry No. 46R 

MR: Accident

DC1: Pneumonia

DC2: Subdural haemorrhage from fall

DC3: –

ABVIs: –

PAs: –

2) Terry No. 393RR

MR: Accident

DC1: Pneumonia

DC2: Femur fracture

DC3: –

ABVIs: –

PAs: –

3) Terry No. 1023

MR: Car accident

DC1: –

DC2: –

DC3: –

ABVIs: –

PAs: –

4) Terry No. 1071R

MR: Accident

DC1: Endocarditis

DC2: Stenosis, hip fracture

DC3: –

ABVIs: –

PAs: –

5) Terry No. 142R

MR: Pneumonia

DC1: –

DC2: Fractured hip

DC3: Senility

ABVIs: –

PAs: –

6) Terry No. 903R

MR: –

DC1: Subdural haemorrhage from fall

DC2: –

DC3: –

ABVIs: –

PAs: –

7) Terry No. 957R

MR: Femur fracture

DC1: –

DC2: –

DC3: –

ABVIs: –

PAs: –

In the original NTB group, the morgue record and/or death certificate stated an accident as the cause of death in four cases (Terry No. 46R, 393RR, 1023, and 1071R). In three additional cases (Terry No.142R, 903R, and 957R), fall, and hip or femur fracture were recorded as the cause of death – these could also be considered as accidents. Unfortunately, as Reviewer 1 already predicted, the number of individuals in the “NTB-accident” group (n=7) would be too low compared to the number of individuals in the TB group (n=234) and in the NTB group without these seven individuals (n=186) for appropriate statistical analysis. Nevertheless, as it can be seen above, none of the seven individuals died of an accident exhibited ABVIs or PAs on the inner surface of the skull. On the other hand, we agree with Reviewer 1 that individuals died of an accident could still have some underlying TB infection or NTB disease associated with meningeal infection. Nevertheless, this is also true for individuals died of NTB disease not associated with meningeal infection. Therefore, we think that very likely, the “NTB-accident” group would not represent a better “control” group for the goals of our paper.

2) Reviewer 1 asked if any of the individuals in the Terry Collection were specifically identified as having died of meningococcal disease (Neisseria meningitidis infection) or other diseases associated with meningeal infections. Following Reviewer 1’s request, we checked all the individuals, and found five cases with the aforementioned conditions as the recorded cause of death: Terry No. 734RR (60-year-old, female) – meningococcal meningitis, Terry No. 455 (25-year-old, female) and Terry No. 719 (25-year-old, male) – luetic meningitis, Terry No. 530 (57-year-old, female) – streptococcal meningitis, and Terry No. 1284R (62-year-old, male) – pneumococcal meningitis. Unfortunately, none of these individuals were included into the NTB group; therefore, they were not evaluated regarding the presence of ABVIs and PAs. (The individuals composing the NTB group were selected randomly by using a random number generator.)

3) As for lines 228–231, Reviewer 1 asked us to give in parentheses the number of individuals with ABVIs dying of cardiovascular problems, respiratory diseases, syphilis, etc. We agree with Reviewer 1 that it would be easier for the readers if they would not have to go to the supplementary table to find this information, and we modified the main text accordingly. 

“In the NTB group, the most frequently registered NTB causes of death were cardiovascular problems (six cases), followed by respiratory diseases (three cases), syphilis (two cases), and different types of cancer (two cases) among individuals exhibiting ABVIs on the inner surface of the skull – in several cases, more than one of these conditions were recorded as the cause of death (S2 Table).”

4) Reviewer 1 noted that in the figures, it would be helpful if we would add small arrows to draw the reader’s attention specifically to the features we wish to highlight. We agree with Reviewer 1 and added the requested arrows in Figures 1, 3, and 5.

In the revised version of our manuscript, we tried to execute all suggestions of the reviewer. I hope this new version will be suitable for publication in PLOS ONE. 

Thank you again for the reviewer’s insightful and constructive comments and your editorial work!

Sincerely yours,

Dr. Olga Spekker, PhD

Postdoctoral researcher

Department of Biological Anthropology, University of Szeged

Közép fasor 52, H-6726 Szeged, Hungary

Email: olga.spekker@gmail.com

Tel: +36 20 807 72 94

---

## [Editor Report · Decision Letter 1]

18 Aug 2020

Tracking down the White Plague. Chapter two: The role of endocranial abnormal blood vessel impressions and periosteal appositions in the paleopathological diagnosis of tuberculous meningitis

PONE-D-20-18111R1

Dear Dr. Spekker,

We’re pleased to inform you that your manuscript has been judged scientifically suitable for publication and will be formally accepted for publication once it meets all outstanding technical requirements.

Kind regards,

JJ Cray Jr., Ph.D.

Academic Editor

PLOS ONE
---

## [Editor Report · Acceptance letter]

20 Aug 2020

PONE-D-20-18111R1 

Tracking down the White Plague. Chapter two: The role of endocranial abnormal blood vessel impressions and periosteal appositions in the paleopathological diagnosis of tuberculous meningitis 

Dear Dr. Spekker:

I'm pleased to inform you that your manuscript has been deemed suitable for publication in PLOS ONE. Congratulations! Your manuscript is now with our production department. 

Kind regards, 

on behalf of

Dr. JJ Cray Jr. 

Academic Editor

PLOS ONE